# Influence of COVID-19 Pandemic Restrictions on College Students’ Dietary Quality and Experience of the Food Environment

**DOI:** 10.3390/nu13082790

**Published:** 2021-08-14

**Authors:** Francilia Brito Silva, Dawn E. Osborn, Meghan R. Owens, Tracie Kirkland, Carolyn E. Moore, Mindy A. Patterson, Wesley J. Tucker, Derek C. Miketinas, Kathleen E. Davis

**Affiliations:** 1Department of Nutrition and Food Sciences, Texas Woman’s University, Denton, TX 76024, USA; FBritosilva@twu.edu (F.B.S.); DOsborn@twu.edu (D.E.O.); 2Department of Nutrition and Food Sciences, Texas Woman’s University, Houston, TX 77030, USA; MOwens14@twu.edu (M.R.O.); CMoore8@twu.edu (C.E.M.); MPatterson14@twu.edu (M.A.P.); WTucker1@twu.edu (W.J.T.); DMiketinas@twu.edu (D.C.M.); 3Department of Nursing, University of Southern California, Los Angeles, CA 90089, USA; TKirkland4@twu.edu; 4Institute for Women’s Health, College of Health Sciences, Houston, TX 77030, USA

**Keywords:** food security, healthy eating index, dietary quality, coronavirus, qualitative assessment, shopping habits, food preparation

## Abstract

The COVID-19 pandemic restrictions sent college students online and off campus, potentially reducing access to healthy food. The objective of this cross-sectional, internet-based study was to use qualitative and quantitative survey methods to evaluate whether COVID-19 pandemic restrictions in Texas, USA affected college students’ ability to buy food, how/what they shopped for, how they prepared food, what they ate, how they felt about eating, and overall dietary quality (assessed using Healthy Eating Index [HEI] scores). Survey responses from 502 students (87.5% female; 59.6% nonwhite, mean age 27.5 ± 0.4 years, >50% graduate students) were analyzed. The qualitative analysis of open-ended questions revealed 110 codes, 17 subthemes, and six themes. Almost all students experienced changes in at least one area, the most common being changes in shopping habits. Participants with low or very low food security had lower HEI scores compared to food secure students (*p* = 0.047). Black students were more likely to report changes in their ability to buy food (*p* = 0.035). The COVID-19 restrictions varied in their impact on students’ ability to access sufficient healthy food, with some students severely affected. Thus, universities should establish procedures for responding to emergencies, including identifying at-risk students and mobilizing emergency funds and/or food assistance.

## 1. Introduction

The COVID-19 pandemic has had major deleterious health and economic impacts on the United States (U.S.) and the rest of the world. In the early stages of the pandemic in the U.S. (March–June 2020), local and state-wide mandates (e.g., stay-at-home measures and business closures) and the pandemic itself resulted in large increases in unemployment [1] and food insecurity [2]. In the U.S., university campuses closed, sending many students home to continue their studies remotely. At the same time, interruptions in food supply chains [3] and job layoffs in the service industry [4] negatively affected some college students’ and their families’ ability to buy healthful food [4,5]. Prior to the pandemic, college students in the U.S. were already an at-risk population for food insecurity, with ~43% of students reporting food insecurity [6,7] compared to 10.5% in the general U.S. population in 2019 [8]. Thus, understanding the impact of the early COVID-19 pandemic restrictions on college students’ ability to purchase and consume healthy food (dietary quality) is an important public health issue.

Many studies have evaluated the impact of the early COVID-19 pandemic on dietary quality or other aspects of dietary intake [9,10,11,12,13,14,15,16,17,18,19,20,21,22,23,24,25,26,27,28,29,30,31]. The majority of studies reported lower dietary quality [9,11,14,15,19,20,21,22,29,30,31], with one study showing improved overall HEI [16] and others indicating improved dietary quality in some aspects [20,21,24,27], such as increased intake of fruits and vegetables [27] and decreased intake of alcoholic beverages [20,24,27] and sugar-sweetened beverages [21]. Four of the aforementioned studies evaluated the effect of the pandemic on dietary intake among college students, including one in Vietnam [15], one in Italy [13], one in Spain [27], and one multi-country study, which included a small sample from the U.S. [11]. However, no study to date has evaluated the impact of the COVID-19 pandemic on university students’ shopping habits, meal preparation practices, or how they feel about eating. In addition, prior studies have had little consistency in how they have evaluated dietary changes and diet quality, with only a few using reliable indicators of dietary quality such as HEI [16,31]. The purpose of this study was first to evaluate whether the early COVID-19 pandemic affected Texas Woman’s University (TWU) students’ ability to buy food, how/what they shopped for, how they prepared food, what they ate, and how they felt about eating, identifying key themes in any changes using qualitative methods. Second, dietary quality was assessed using both quantitative (HEI) and qualitative methods. Finally, this study sought to evaluate whether demographic factors were associated with qualitative findings or HEI.

## 2. Materials and Methods

### 2.1. Study Design and Participants

This was a cross-sectional study of college students using an online survey administered in PsychData (PsychData LLC, State College, PA, USA). All study procedures were approved by the Institutional Review Board for Human Subjects Research at TWU (Houston, TX, USA) (IRB#: FY2020-314). The complete methodology has been described elsewhere [32]. Between 12 May and 12 June 2020, students on all three campuses of this large, diverse, state-funded university in Texas, USA, were invited via email to participate in an online survey. The university has three campuses in the same state, two in large, metropolitan areas (Houston and Dallas, TX, USA) and one in a suburban area (Denton, TX, USA). In 2020, TWU had a total student enrollment of 14,888 students across three campuses (88% female, 38% graduate, 62% undergraduate, 40% white, 27% Hispanic, 18% black, 11% Asian/Pacific Islander) [32].

Eligibility criteria included being 18 years of age or older, speaking English fluently, and having internet access. A student listserv was used to disseminate the survey to all students in a weekly email. Students who clicked on the survey link were directed to the survey information page, which described the study purpose, risks, and benefits. Students consented to participate by clicking “next” to accept the informed consent. The online survey took about 15 min to complete.

### 2.2. Survey Tool

The survey had three sections: (1) food security assessment using the validated 2-item food sufficiency screener [33] and U.S. Department of Agriculture (USDA) Food Security Survey Module (FSSM): six-item short form [34] (Results previously reported [32]); (2) qualitative assessment of the impact of the COVID-19 pandemic on students’ financial ability to purchase food, shopping, meal preparation, diet, and feelings regarding eating (See Table 1 for a list of the qualitative questions); (3) sociodemographic characteristics including asking about a participant’s height and weight, age, sex, race/ethnicity, college undergraduate or graduate status, campus location, and current participation in federal food assistance programs (SNAP and WIC). Upon completion of the survey, students were directed to enter their email address for a chance to win one of 25, 50 USD digital gift cards.

### 2.3. Assessment of Dietary Quality

All participants who entered their email address at the end of the initial online survey received a link to a follow-up survey that assessed habitual dietary intake using the Automated Self-Administered 24-h (ASA24) Dietary Assessment Tool, version 2020, developed by the National Cancer Institute (Bethesda, MD, USA). The ASA24 prompts participants to record everything consumed in the previous 24 h from midnight to midnight using detailed probes and has been validated against interviewer-administered 24-h recalls, food records, and actual intakes [35,36,37]. Total and component HEI-2015 scores were derived from the results of ASA24 dietary recalls using an SAS code that is outlined on the National Institutes of Health website [38]. HEI score is a validated measure of dietary quality that reflects how well an individual’s dietary intake aligns with the U.S. Dietary Guidelines for Americans. The HEI total score is comprised of thirteen component scores for food and nutrient adequacy: total vegetables, greens and beans, total fruit, whole fruit, whole grains, total dairy, total protein, seafood and plant protein, and fatty acid ratio; and food and nutrient moderation: sodium, refined grains, saturated fatty acids, and added sugar. The component scores are added to calculate the total HEI score, which ranges from 0 to 100 [39].

### 2.4. Data Analysis

The data set was imported into NVivo 12 (QSR International, Burlington, MA, USA) for general, qualitative thematic analysis. To decrease workload and promote reliability of the analysis, three coders (F.B.S., D.E.O., K.E.D.) initially met to discuss the analytic approach and research questions. To limit bias, each coder first, independently, coded five participants’ responses, then they met to compare codebooks and discuss the responses where meaning was unclear and come to consensus on the initial codes and how to code such responses. Recoding was performed. Then, the three coders coded ten more participant responses in common and met again. Recoding was performed again, and 35 more participants’ data were coded in common. After these 50 participants’ data were coded in common, the three coders discussed the codes and their definitions in-depth and reached consensus on the codes. After the initial three meetings described above, coding proceeded for the remaining 452 participants with each coder independently coding about 150 responses each. During this process, they communicated about new potential codes and questions about the correct meaning of existing codes by using a Google doc codebook with descriptions of the codes and illustrative quotes. When questions arose, coders would notify the other coders via the Google doc so that they could respond and come to consensus. Finally, the coders met to review the codes, dropped codes with only one response, and developed subthemes and themes. The coders also discussed each code, including how many responses they had for each. In this process, if one coder determined they had not properly understood or given attention to a code, recoding was performed. Then one final meeting to evaluate the project was conducted. Finally, codes, subthemes, and themes reported by each participant were exported to an Excel file for comparison of themes according to demographic factors and food security status.

Responses were weighted according to sex and race/ethnicity demographics as described previously [32] and were adjusted for nonresponse for those who completed the dietary recall. Weighted mean HEI-2015 scores were compared within each theme; for example, those whose responses were classified within Theme 1 versus those whose responses were not classified within Theme 1. HEI-2015 scores were also compared across food security status. Continuous variables are reported as mean ± SEM unless otherwise indicated. Independent samples *t*-tests and Rao−Scott Chi-Square ($\chi {²}_{RS}$) tests were used to test for differences between continuous and categorical variables, respectively. A *p*-value < 0.05 was considered statistically significant. All analyses were performed using NVivo 12 (QSR International, Burlington, MA, USA) and SAS software version 9.4 (SAS Institute Inc., Cary, NC, USA).

## 3. Results

Six-hundred fifty-one students responded to the survey from a total population of 14,888 students among the three campuses (response rate 4.4%). After excluding 137 responses due to incomplete responses and 12 duplicates, 502 participants’ qualitative responses were included. Valid responses were available for 301 participants to include in the dietary quality analysis. TWU is a large, public university attended primarily by women; thus, the sample was 87.5% female. Racial/ethnic composition was as follows: 40.4% white, 18% black, 26.9% Hispanic, 10.3% Asian, 4.4% other. Over half of the students who responded were graduate students. The mean age was 27.5 ± 0.4 years. As previously reported, 34.5% of the participants were classified as food insecure [32]. Additional details about the participants’ demographics may be found in Owens et al. [32].

Qualitative analysis of the open-ended survey questions revealed 110 codes, 17 subthemes, and six themes: (1) no effects of COVID-19 pandemic; (2) changes in financial ability to buy food; (3) changes in shopping habits; (4) increased cleaning; (5) changes in food preparation and food/drink consumption; and (6) changes in attitudes, feelings, and habits related to food/eating (See Table 2 for a complete list of themes, subthemes, and codes and Table 3 for illustrative quotes for each theme).

### 3.1. No Effects of COVID-19 Pandemic

It was more common for participants to report changes in at least one area than no change due to COVID-19. Participants indicating no effect included 109 indicating “no financial challenges”, 168 indicating “enough funds for food”, 101 indicating “no change in diet”, 121 indicating “no change in feelings about eating”, 114 indicating “no change in food preparation”, and 28 indicating “no change in shopping.” As indicated above, financial challenges were common, but having enough funds for food was also often reported. Reporting “no effect of COVID-19” did not differ according to age group and class status (freshmen, sophomores, graduate students, etc.). However, students with high or marginal food security status were more likely to report no effect of COVID-19 compared to students with low or very low food security ($\chi {²}_{RS}$ = 38.1; *p* < 0.001).

### 3.2. Changes in Financial Ability to Buy Food

The most common code within the “financial difficulties” subtheme was “limited funds for food” (*n* = 88), followed by “personal financial challenges” (*n* = 62). Participants endorsing “limited funds for food” made comments such as “Until my unemployment began, i was unable to purchase groceries. when i could purchase groceries, i couldnt always afford the food remaining, and i bought what i would. (sic)”.

The “food assistance” subtheme (participants noting use of food assistance programs) was noted the least often, with two codes reported by 15 participants (Table 3). Slightly more common were the “food expenses” and “work limitations” subthemes with three codes reported by 83 participants for each subtheme. The most common “food expenses” code was “increased food prices” (*n* = 31). The most common “work limitations” code was “lost job” (*n* = 53). The “work limitations” subtheme described a variety of situations that arose from shutdowns or reduced hours in businesses. Families experienced job losses differently, with some families reporting not all family members losing jobs (“I have lost my job due to the Coronavirus, however thankfully my husband is still employed and we are able to have enough money”) and other families experiencing greater impacts (“No one in my family is working right now because of coronavirus pandemic. It is affecting us in many ways including financial ability to purchase food. We are barely surviving because of our financial situation”).

Students who reported employment changes were more likely to provide responses within this category (57.9% vs. 41.9%; *p* = 0.015). Black students were also more likely to report changes in this theme (*p* = 0.035). Black students’ responses comprised 21.1% of the responses classified within this theme versus 7.9% of responses that were not classified within this theme.

### 3.3. Changes in Shopping Habits

The area in which change was an almost universal finding was shopping, with only 28 participants reporting no change in shopping. Male students were less likely to report changes in this theme (8.5% were classified within this theme vs. 21.5% who were not classified within this theme; *p* = 0.025). Changes in shopping habits were classified into three subthemes: “change in shopping time or frequency”, “changes in usual store”, and “food accessibility challenges”. The “change in shopping time or frequency” subtheme included three codes, with the most common being “shopping less often” (*n* = 137). Reasons for shopping less often included reducing risk of COVID-19 exposure, fear of exposure, and a lower food budget. Two illustrative quotes that depict these subthemes well are “COVID-19 has created fear within us to buy food or going out to buy essentials” versus “One of the easiest ways to cut down costs is to be more conscious of what I buy and try to go every other week instead of once a week”.

The “food accessibility challenges” subtheme was the most common within this theme, with six codes reported by 293 participants. The most common code was “food shortage at stores” (*n* = 174, more than one-third of participants). Participants reported several issues related to food shortages, including trouble finding food staples: “we were unable to get the things we normally ate and limited shopping to once a week, but had to go to multiple stores just to get the staples of our household”. Other participants reported particular accessibility challenges such as “purchasing less meat” (*n* = 33) due to increased cost or lack of availability.

### 3.4. Increased Cleaning

One code, “increased cleaning” (*n* = 35) was also classified as a theme because of the wide variety of responses in participants’ descriptions of “cleaning”. Sometimes increased cleaning was reported relating to shopping, sometimes cooking, and occasionally for other reasons: “We are constantly cleaning the counters and washing everything”. “Once I bring in all groceries, everything is sprayed with Lysol. My fresh fruits and vegetables are washed and soaked in hot water to try and sanitize them as much as possible before consumption”. “I wear a mask, I wipe down my cart, and I keep hand sanitizer with me”.

### 3.5. Changes in Food Preparation and Food or Drink Consumption

Many changes in both food preparation methods and food and drink consumption were reported. Various reasons for these changes included cost, loss of income, purchasing nonperishable foods to avoid going to the store as often, and working more and thus not having time to cook. This theme included five subthemes (Table 2). One of the most common subthemes was “changes in type of food eaten” with 15 codes and 441 participants endorsing codes in this subtheme, some indicating potentially improved diet quality (“increased fruit and vegetable intake” *n* = 27) and others indicating a decline in diet quality (“buying or eating more convenience foods” *n* = 87, which was also the most common code in the subtheme). “We just eat cheaper, more processed foods”. The second most common code in this category was “changing typical foods for cheaper foods” (*n* = 60).

Cooking was another common subtheme, with eight codes and 291 participants with responses in this area. The most common was “cooking more meals at home” (*n* = 144). Participants mentioned eating out less and cooking more meals at home: “Due to the pandemic, my family and I have relied on only cooking home meals and have reduced dramatically our intake of fast food”.

In the subtheme “managing food availability at home”, the most common code was “food stretching” (*n* = 51) with participants describing practices to make food last longer: “Because of the virus i batch cook more because it makes my food stretch further” (sic), and others reporting reducing intake: “We eat less than normal now because we have to make sure we have enough to last throughout the week at least”.

Another very common subtheme was “meal and snack intake and pattern changes” with six codes reported by 401 participants. Codes and responses showed that the pandemic affected students differently, but most still experienced changes in dietary habits. Some students experienced reduced intake of food, with the most common code being “eating less” (*n* = 148). Reasons for eating less included making food last longer to avoid shopping, “changes in routine/schedule and other emotional reasons”, not being able to be as active and intentionally eating less, or even “because of the number of Meet or Zoom meetings I have getting in the way of making a meal”. Others cited “rationing” and not having enough money for food. The second most common code in this subtheme involved the opposite type of behavior: “increased snacking” (*n* = 78). Common types of quotes indicated “I do tend to snack more because I am working from home”.

“Restaurant and food habits” was the last subtheme in this category with three codes and 233 participants reporting. The most common code was “limiting take out and eating out” (*n* = 162): “I have not purchased prepared food from any restaurants since stay-at-home orders were placed 13 March 2019. I do not feel comfortable consuming any food prepared outside of my home”. However, the second most common code in this subtheme (*n* = 43) was “increased take out or eating out”.

### 3.6. Changes in Attitudes, Feelings, and Habits Related to Food or Eating

Within this theme, there were five subthemes. Students with low or very low food security reported changes in this theme more often (*p* = 0.015). The “changes in attitudes towards food” subtheme included six codes with 107 participants endorsing change. A common code in this subtheme was “appreciating or enjoying food more” (*n* = 30). One student reported, “I think I enjoy it more now because I make my food how I like it and I have more time to enjoy it instead of eating in front of my computer working”. A minority of references reported “appreciating or enjoying food less” (*n* = 13). However, another code in this subtheme was “less desire to eat” (*n* = 32). Quotations in this code were diverse, ranging from “I feel like I can go days without eating. I still feel fat even when I barely eat,” with a focus on body image and “I eat for function and for necessity now, not for pleasure”, with several quotations reporting a similar focus on food to survive.

The subthemes “changes in feelings about food (increased positive feelings)” and “changes in feelings about food (increased negative feelings)” included four codes reported by 75 participants and 12 codes reported by 180 participants, respectively. That is, increased negative feelings were reported more than twice as often as increased positive feelings. The most common “positive feelings” code was “feeling more thoughtful about eating or drinking” (*n* = 31): “I notice how I feel after eating, which is a huge change”. The two most common “negative feelings” codes were “missing foods used to eat before COVID” (*n* = 32) and “fear of contamination or exposure to COVID” (*n* = 33). Some described feeling uncomfortable consuming food prepared outside the home or worry about going to the grocery store. The final subtheme in this theme was “disordered eating changes”. The most common code was “emotional, boredom, or stress eating” (*n* = 61). A common type of response was “I feel much more out of control and stressed about my eating habits”.

### 3.7. Dietary Quality

Mean total HEI-2015 score was 53.8 ± 0.9 overall. Students who endorsed changes in shopping habits had higher HEI scores compared to those with no change (55.1 ± 1.0 versus 50.4 ± 1.8; *p* = 0.024). In addition, those with very low food security had lower HEI scores compared to those with high or marginal food security (49.6 ± 2.5 versus 54.9 ± 1.0; *p* = 0.047).

The average whole grain component score was the lowest of the 13 components for the total sample (3.1 ± 0.2). Those whose responses were not classified into the “Changes in food preparation and food or drink consumption” theme had significantly lower mean whole grain component scores than those whose responses were classified into this theme (1.7 ± 0.5 vs. 3.3 ± 0.3; *p* = 0.009). Table 4 presents the HEI 2015 component and total scores for the total sample and within each theme.

## 4. Discussion

This cross-sectional, internet-based study explored the influence of the early COVID-19 pandemic restrictions on TWU college students’ ability to buy food, shopping habits, food preparation methods, diet, feelings about food, and dietary quality using both qualitative and quantitative methods. This study found that students experienced the early pandemic in diverse ways. Few students reported no effects of the pandemic, with almost all reporting change in at least one area, with shopping being the area most often affected. Within this theme, food shortages were the aspect most frequently cited. However, while many students reported financial difficulties for themselves or their families, many reported having enough funds for food. Black students were more likely to report changes in employment affecting their ability to buy food. Impacts on food preparation methods were diverse. Students were more likely to report increases in cooking meals at home and decreases in eating out or getting takeout, but this was not universal, with a small group reporting the opposite impact. Similarly, while it was more common to report increases in purchases of convenience foods and a reduced dietary quality, some students reported increased dietary quality. However, quantitative assessment of dietary quality using HEI indicated that the HEI of food-insecure students was lower than the HEI in food-secure students. Regarding attitudes and feelings towards food, negative feelings were more than twice as common among students compared to positive feelings. However, some students reported thinking more about their food and enjoying it more. To summarize, the COVID-19 pandemic affected students differently based on whether they had lost a job or experienced other financial impacts and whether they had a safety net (such as other family members or roommates still working). In addition, black students were more likely to have insufficient funds for food.

This study was not unique in its finding of grocery shopping as an area strongly affected by COVID-19 and its restrictions; the COVID-19 pandemic caused changes in shopping habits among people around the world. A multi-country survey conducted during the pandemic showed that people tended to buy food in bulk and reported higher prices of food staples [28]. Skotnicka et al. [25] found that Slovenian adults tended to shop for food less frequently and eat out or order takeout more often. In the current study, while many U.S. college students reported shopping for food less frequently, they also reported cooking more and eating out less, which contrasts with the prior study of Slovenian adults [25]. In a prepandemic study of Italian college students’ shopping habits, students reported that they liked to shop with friends and spend time in grocery stores to choose products based on price and label information [40]. In addition, students reported enjoying eating lunch out with friends [40]. However, in the current study, students often reported changing their shopping habits to shop less frequently, reduce the amount of time spent in the store during grocery shopping, and have only one person in each household assigned to grocery store shopping to prevent risk of COVID-19 infection.

A limited number of qualitative studies have been conducted investigating dietary intake during the COVID-19 pandemic [23,28,41,42]. Sansah et al. [42] recorded qualitative responses of 27 Ghanaian adults during COVID-19 lockdown and found that some individuals reported trying to eat more healthfully, while others reported boredom and poor diet [42]. However, other studies have found changes in attitudes towards eating, eating habits, and overall dietary intake using various survey assessments [23,28,41]. For example, Buckland et al. [23] reported an increased intake of energy dense foods and difficulty controlling cravings. Similarly, a study of Saudi women reported that 53% of women experienced moderate or high “emotional eating” during the COVID-19 pandemic [41]. In the present study, 61 (12%) participants reported emotional, boredom, or stress eating. In addition, qualitative responses coded for reduced dietary quality were found in only 65 (13%) of the 502 participants compared to 32 (6%) participant responses being coded to “improved quality diet”, indicating twice as many participants endorsing negative versus positive changes in dietary quality. These findings are similar to a multi-country study of adults conducted by Jafri et al. [28], which found that adults reported consuming fewer preferred foods, reducing portion sizes, and/or reducing the number of meals during the COVID-19 pandemic.

The quantitative analysis of dietary quality in this study found that HEI scores were lower (53.8 in the current study) compared to the U.S. national average of 58.3 (adults 19 to 64 years old) reported prior to the pandemic [43]. In addition, the total HEI scores in the present study were much lower than a 2019 study of U.S. college students at a state university, who reported a mean total HEI score of 63.3 [44]. A limited number of studies have quantitatively assessed HEI during the COVID-19 pandemic [16,31]. In a sample of adults in Quebec, Canada, LaMarche et al. [16] observed a small increase in 2015-HEI scores (+1.1 points) during the early COVID-19 lockdown relative to the prepandemic baseline (mean: 69.0 points). In contrast, Bogataj Jontez et al. [31] found a decrease in the HEI score from 64.6 at baseline to 61.6 during a 2-month lockdown in Slovenian adults, with HEI scores returning to near baseline after lockdown. Compared to these prior studies [16,31], the college students in the present study had much lower HEI scores. However, since our study did not include a prepandemic baseline for HEI, it is unclear whether the low HEI scores are the result of the COVID-19 pandemic.

Several previous studies assessed dietary quality using other survey assessments during the COVID-19 pandemic [9,12,14,17,18,19,21], with a wide range in respondent experiences reported. Some populations reported adverse changes in diet quality and dietary behaviors including greater energy intake [9,12,17], higher consumption of unhealthy foods [14], and increased snacking frequency [18]. An Italian study of adolescents and adults found that over one-third of their sample reported consuming a less healthy diet during the early lockdown. However, in contrast, one-third of their sample reported consuming a healthier diet [12]. In the present study, the differences in responses were especially evident in the qualitative data, with 168 (33%) respondents reporting enough funds for food and 101 (20%) respondents endorsing no change in diet; whereas 88 (18%) respondents reported limited funds for food and many mentioned some type of dietary change. Other studies have focused less on differences within populations, instead, focusing on overall trends in dietary quality or dietary behaviors. A multi-country study of adults that used the Short Diet Behaviors Questionnaire for Lockdowns to assess diet found higher levels of reporting “consuming unhealthy food” and a lower score on diet behaviors during home confinement [9]. Similarly, studies in Spanish [14] and French [17] adults reported increased energy intake [14] and lower overall dietary quality [14,17]. Other studies have reported on particular aspects of the diet rather than overall diet quality during the pandemic. Two studies reported reduced intake of protein [21,24] whereas in this study 33 (7%) participants reporting purchasing less meat in response to the question about changes in shopping habits, usually related to shortages or cost increases. In addition, a reduced intake of fresh produce has also been reported in some studies [19,21]. In the present study, just 38 (8%) participants reported reduced fruit and vegetable intake when asked about changes in their food preparation practices and diet. However, it should be noted that in the present study, because of its qualitative nature, greater numbers of students may have experienced a particular change, but only a subset may have found it important enough to mention in answer to a question. Finally, there is also some evidence of an increase in potentially adverse dietary behaviors during the pandemic. In a large sample of adults (*n* = 2002) in the United Kingdom, a large number of participants reported negative changes in eating behaviors, including increased snacking frequency and less control around food [18], whereas in the present study 61 (12%) participants mentioned similar negative changes (“emotional, boredom, or stress eating”).

To date, only three studies have assessed dietary intake during the COVID-19 pandemic in college students [11,13,15]. Amatori et al. [13] surveyed college students in Italy about dietary habits during COVID-19 home isolation. However, they did not report dietary quality, only its relationship with exercise and mood, making comparisons impossible. Duong et al. [15] reported that 43% of nursing and medical students surveyed in Vietnam had healthier eating behavior during the pandemic, although the majority reported no change or worse eating behavior (combined response types). These changes were assessed using a self-rated approach, with students being asked to simply rate their eating behavior as less healthy, unchanged, or healthier [15]. Finally, a multi-country study assessed college students’ health behaviors (including a sample of 1280 U.S. students) using the Starting the Conversation questionnaire, an eight-item simplified food frequency questionnaire to assess “dietary risk” based on common food patterns [11]. Dietary risk scores were relatively consistent across the seven countries; however, the U.S. students had the highest absolute mean dietary risk scores [11]. In addition, similar to the study by Duong et al. [15], Du et al. [11] also asked students to self-rate their diet during the COVID-19 quarantine compared to prepandemic, with choices including “healthier than before”, “less healthy than before”, or “did not change”. One in three U.S. students reported eating less healthfully during the COVID- 19 pandemic [11]. In the current study, 65 (13%) students generally or specifically reported consuming a lower diet quality during the pandemic. Furthermore, a quantitative analysis of dietary quality in this study found that HEI scores were lower than the U.S. national average [43] and those of U.S. college students [44] reported prior to the pandemic.

Food security plays an important role in diet quality, with food-secure individuals typically exhibiting higher diet quality than food-insecure individuals [45]. Indeed, in the current study, students with very low food security had HEI scores that were more than five points lower than students with high or marginal food security. In addition, black students were more likely to report changes in the theme “changes in financial ability to buy food”. The qualitative responses made it clear that some students were experiencing severe disruptions in employment, income, and dietary intake while others were enjoying time off and experimenting with new foods and cooking at home. Social determinants of health (SDOH) are “conditions in the environments where people are born, live, learn, work, play, worship, and age that affect a wide range of health, functioning, and quality-of-life outcomes and risks” [46]. Among the domains of SDOH are economic stability and education access and quality [46]. While this study’s objective was to explore whether the COVID-19 pandemic restrictions influenced college students’ ability to buy food, shopping habits, food preparation methods, diet, feelings about food, and dietary quality, in the process we found that one of the key domains of SDOH—economic stability—was affected very differently among students, with black students exhibiting higher risk for not having sufficient funds for food during the pandemic. This is consistent with the food insecurity literature in college students prior to the pandemic, with multiple studies showing that black and Hispanic college students were more likely to be food insecure than white students [32,47,48,49,50]. This is important because food insecurity has been shown to be associated with poor mental health [47,48] and worse academic performance [48,51,52] in college students. Stress surrounding one’s financial stability and not having enough food to eat will, unquestionably, negatively affect students’ ability to participate successfully in the educational experience. In the present study, SDOH (particularly economic stability) may (in part) be responsible for the heterogenous responses to qualitative questions related to food security and diet quality. While all students were impacted by COVID-19 restrictions, the differences in access to technological and economic support (safety nets) may have made it more difficult for students with less resources to adapt to online learning and resulted in reduced access to healthy foods.

The strengths of this study included using both qualitative and quantitative methods to assess not just diet, but the ability to buy food, shopping, food preparation, and feelings about food. In addition, this was the first study to use the validated ASA24 tool to assess dietary intake and diet quality in college students during the early stages of the COVID-19 pandemic. The qualitative analysis both supported the findings of the ASA24 and provided insight into the specific issues students were facing, such as food insecurity, access to healthy food, changes in shopping behavior, and changes in their attitudes towards food. Finally, this survey sample was racially diverse and included students from all three campuses located in Texas’s two largest metroplexes (Dallas−Fort Worth and Houston). This is important because minority college students have been shown to be at an increased risk for food insecurity relative to white students [32,47,48,49,50] and the findings in the current study further highlight these public health disparities.

Study weaknesses included poor response rates, probably in part due to the stressors students experienced during the early COVID-19 pandemic associated with campus closures and a rapid shift to online learning. However, this sample, despite including a higher proportion of graduate students compared to the overall student body, was similar in race/ethnicity and gender to the overall TWU population. Furthermore, low survey response rates are unfortunately common in nutrition research in college students [53,54]. Finally, these findings may not be generalizable to college students across the U.S. and in other countries as our sample is limited to one primarily female, diverse, state-funded university in Texas, USA.

## 5. Conclusions

COVID-19 restrictions varied in their impact on college students’ ability to access sufficient healthy food, with some students severely affected while others experienced little to no change. Few students reported no effects of COVID-19 in any area with almost all students reporting at least one change among the investigated areas, the most common being changes in shopping habits. In addition, black students were most likely to experience limitations in their ability to buy food, and food-insecure students were most likely to report low dietary quality (assessed by HEI score).

Our findings suggest that universities need to be better prepared to help students respond to emergency situations with immediate nutrition assistance for at-risk students. While pandemics have been uncommon in recent history (except for COVID-19), many experts predict that the prevalence of pandemics and natural disasters related to climate change will increase in the future [55]. Thus, having mechanisms to identify at-risk students for immediate access to emergency funds and/or nutrition assistance may help students cope better and achieve better academic success during future crises. Furthermore, technology, including text-messaging, use of university social media, virtual nutrition counseling, and other approaches may prove useful in reaching at-risk students. In addition, universities may need to consider building interdisciplinary teams to address this type of planning and advocate for policies to provide resources for students and their families as a part of emergency preparedness for natural disasters and/or unforeseen public health crises.

## Figures and Tables

**Table 1 nutrients-13-02790-t001:** Qualitative Survey Questions.

Please share in what ways (if any) the Coronavirus (COVID-19) pandemic has affected your financial ability to buy food.
Please share how you believe the Coronavirus (COVID-19) pandemic has affected how, where, or when you shop for or purchase food and how it has affected what foods you buy and eat. (These could include grocery shopping or buying foods from other places, such as restaurants.)
Please share any changes you have noticed in how you prepare and/or cook your meals.
Please share any changes you have noticed in how often you eat, what you eat, or how much you eat.
Please share any changes you have noticed in how you feel regarding eating.

**Table 2 nutrients-13-02790-t002:** Qualitative Codes, Subthemes, and Themes of College Students’ Food and Shopping Habits during the Early COVID-19 Pandemic, Including Number and Percentage of Participants Reporting Codes.

Themes	Subthemes	Codes
No effects of COVID-19	Others	Enough funds for food (*n* = 168; 33%) No change in feelings about eating (*n* = 121; 24%) No change in food preparation (*n* = 114; 23%) No financial challenges (*n* = 109; 22%) No change in diet (*n* = 101; 20%) No change in shopping (*n* = 28; 6%)
Changes in financial ability to buy food	Financial difficulties	Limited funds for food (*n* = 88; 18%) Personal financial challenges (*n* = 62; 12%) Family financial challenges (*n* = 49; 10%) Change in living situation (*n* = 14; 3%) Relying on savings (*n* = 8; 2%) Borrowing money for food (*n* = 8; 2%) Supporting other family members financially (*n* = 4; <1%)
	Food assistance	Food donation or food pantry use (*n* = 10; 2%) Relying on SNAP ^1^ (*n* = 5; 1%)
	Food expenses	Increased food prices (*n* = 31; 6%) Spending more on food to stock up (*n* = 28; 6%) ^2^ Spending more on food (*n* = 24; 5%) ^3^
	Work limitations	Lost job (*n* = 53; 11%) Being furloughed (*n* = 7; 1%) Reduced hours at work (*n* = 23; 5%)
Changes in shopping habits	Change in shopping time or frequency	Shopping less often (*n* = 137; 27%) Change in shopping time (*n* = 55; 11%) Visiting more stores to do shopping (*n* = 25; 5%) Shopping more often (*n* = 13; 3%)
	Changes in usual store	Changing stores for cheaper options (*n* = 23; 5%) Changing stores due to lines or crowding (*n* = 16; 3%) Changing stores due to selection available (*n* = 17; 3%) Cheaper delivery service (*n* = 2; <1%)
	Food accessibility challenges	Food shortage at stores (*n* = 174; 35%) Ordering groceries for delivery or curbside pick-up (*n* = 76; 15%) Purchasing less meat (*n* = 33; 7%) Transportation problems (*n* = 4; <1%)
	Others	Shopping more carefully (*n* = 69; 14%) Special diet needs (*n* = 4; <1%)
		Shopping more or buying more due to family members’ return home (*n* = 4; <1%) Shopping related anxiety (*n* = 4; <1%) Increased ride sharing prices (*n* = 2; <1%)
Changes in food preparation or drink consumption	Changes in type of food eaten	Buying or eating more convenience foods (*n* = 87; 17%) Reduced quality diet (*n* = 65; 13%) Changing typical food for cheaper foods (*n* = 60; 12%) Eating less desirable foods (*n* = 52; 10%) Reduced fruit and vegetable intake (*n* = 38; 8%) Improved quality diet (*n* = 32; 6%) Increased fruit and vegetable intake (*n* = 27; 5%) Increased carbohydrate intake (*n* = 22; 4%) Prioritizing healthy foods (*n* = 17; 3%) Buying or eating fewer convenience foods (*n* = 16; 3%) Increased water intake (*n* = 9; 2%) Changing from organic food to conventional food (*n* = 7; 1%) Increased meat intake (*n* = 4; <1%) Changes in alcohol intake (*n* = 3; <1%) Increased fried food intake (*n* = 2; <1%)
	Cooking	Cooking more meals at home (*n* = 144; 29%) Incorporating new ingredients in cooking (*n* = 34; 7%) Cooking healthier meals (*n* = 27; 5%) Less desire to cook (*n* = 20; 4%) Baking more (*n* = 19; 4%) Cooking more cautiously (*n* = 19; 4%) Cooking less (*n* = 18; 4%) ^4^ Cooking less to avoid food waste (*n* = 10; 2%) ^5^
	Managing food availability at home	Food stretching (*n* = 51; 10%) Using leftovers (*n* = 31; 6%) Less food waste (*n* = 23; 5%) Increased meal planning (*n* = 21; 4%) Increased meal prepping (*n* = 19; 4%) Repackaging food for later use (*n* = 10; 2%) Buying less of different items to avoid food waste (*n* = 9; 2%) Eating less or skipping meals to leave food for other family members (*n* = 8; 2%) Growing fruits and vegetables at home to save money or other reasons (*n* = 6; 1%)
	Meal and snack intake and pattern changes	Eating less (*n* = 148; 29%) Increased snacking (*n* = 78; 16%) Eating more (*n* = 63; 13%) Meal skipping (*n* = 50; 10%) Decreased snacking (*n* = 36; 7%) Change in timing of meals or snacks (*n* = 26; 5%)
	Restaurant food habits	Limiting take out and eating out (*n* = 162; 32%) Increased take out or eating out (*n* = 43; 9%) Ordering food to support local business (*n* = 28; 6%)
Changes in attitudes, feelings, and habits related to food or eating	Changes in attitudes towards food	Less desire to eat (*n* = 32; 6%) Appreciating or enjoying food more (*n* = 30; 6%) Increased attention to what and how much eating (*n* = 25; 5%) Appreciating or enjoying food less (*n* = 13; 3%) Conflict with family members over best way to eat (*n* = 4; <1%) Hating what you eat (*n* = 3; <1%)
	Changes in eating habits	Eating less carefully (*n* = 11; 2%) Choosing carefully what to eat (*n* = 8; 2%) Eating more meals as a family (*n* = 7; 1%) Eating outdoors more (*n* = 2; <1%)
	Changes in feelings (increased negative feelings)	Fear of contamination or exposure to COVID-19 (*n* = 33; 7%) Missing foods used to eat before COVID-19 (*n* = 32; 6%) Feeling “fat” or concerned about weight gain (*n* = 22; 4%) Feeling less energized due to eating food perceived to be less healthy (*n* = 21; 4%) Feeling afraid of eating due to food uncertainty (*n* = 18; 4%) Feeling guilty about eating (*n* = 16; 3%) Eating-related anxiety (*n* = 14; 3%) Feeling hungry (*n* = 6; 1%)
		Feeling out of control with eating (*n* = 6; 1%) Feeling afraid of not being able to buy healthy foods (*n* = 4; <1%) Missing eating with others outside household (*n* = 4; <1%) Missing organic and whole foods (*n* = 4; <1%)
	Changes in feelings (increased positive feelings)	Feeling more thoughtful about eating and/or drinking (*n* = 31; 6%) Feeling good about cooking (*n* = 21; 4%) Feeling healthier (*n* = 21; 4%) Improved weight and energy (*n* = 2; <1%)
	Disordered eating changes	Emotional, boredom, or stress eating (*n* = 61; 12%) Less emotional or stress eating (*n* = 9; 2%) Retriggering of eating disorder (*n* = 4; <1%)
Increased cleaning	Other	Increased cleaning (*n* = 35; 7%)

**Notes: ^1^** SNAP: Supplemental Nutrition Assistance Program. ^2^ Spending more on food due to higher prices or reason not specified. ^3^ Spending more on food to stock house with longer-lasting food supplies. ^4^ Cooking less for reasons not specified; sometimes also reported with using more convenience foods. ^5^ Cooking less reported specifically to reduce food waste.

**Table 3 nutrients-13-02790-t003:** Themes and Illustrative Quotes of College Students’ Experiences with Food During the Early COVID-19 Pandemic.

Themes	Illustrative Quotes
No effects of COVID-19	“Our household has not been affected financially. My husband has a stable job with steady income”. “My ability to purchase food has not changed from prior to this pandemic. Luckily, I am still receiving a full paycheck from my job and that is a blessing”. “We are eating the same amount”. “There are not any observable changes in how often, what, and how much we eat”.
Changes in financial ability to buy food	“I have had to rely on others. I have many Bill’s that sometimes I couldn’t buy food I have to spend it on other things such as a car and car insurance so I had to borrow money”. (sic) “Since my family and I lost our jobs, we have been having to go to church food drives once a week to be able to get food since we didn’t have enough money”. “I have to spend more money to buy more food and necessities that are available because you never know if they will be there next time”.
Changes in shopping habits	“Have to shop early in the mornings in order to get the products we need at prices we can afford”. “We now shop with masks and shop on a weekday when it is less crowded. We also are buying more shelf stable foods”. “I shop less frequently and with more awareness of cost and price per ounce”. “I shop at smaller, less occupied grocery stores” “We have been shopping at stores with less expensive grocery options”. “In my family, we were unable to get the things we normally ate and limited shopping to once a week, but had to go to multiple stores just to get the staples of our household”.
Changes in food preparation and food or drink consumption	“I am eating less healthier because of the cost of healthy versus unhealthy foods”. (sic) “I prepare only what I know I’ll eat now, I usually like to prepare leftovers. I also use less ingredients, like not as many spices and herbs as I would like to because I’m not sure when I can get more”. “I do tend to snack more because I am working from home”. “I am meal prepping more and eating lower quality food”. “We eat more often and snack more often since mealtime isn’t structured around lunchtime breaks and schedules”. “We eat when we are hungry instead of when it is ‘time to eat.’” “Lastly, when it comes to purchasing food from restaurants, I have only bought food from out once. Before this pandemic, I would eat out at least 2 times a week”. “Also, we have ordered food to support local businesses and individuals who are not able to work”.
Changes in attitudes, feelings, and habits related to food or eating	“This makes me think about what I am eating”. “More mindful of what I eat, since I can go to the kitchen whenever and sometime find myself eating simply because I can”. “Being in this lockdown situation, my family and I usually have breakfast and dinner together”. “I have been afraid to eat because I was scared we would run out of food”. “I am more willing to cook now and have tried to make it something I enjoy by trying new recipes with the food available. Before the pandemic I did not enjoy cooking at all”. “At the beginning of this pandemic and quarantine, I noticed that I was eating a bit more out of boredom, from staying home all day”.
Increased cleaning	“Only one person in my home does the weekly grocery shopping, and we disinfect and wash EVERYTHING before we put it away or consume it”. “Everything is thoroughly washed and cooked. I do not consume any undercooked meat” “We wash fruits and vegetables as soon as we purchase them rather than wait until we are about to cook it like we did for years”.

**Table 4 nutrients-13-02790-t004:** Healthy Eating Index 2015 Component and Total Scores Across Themes ^1^ in College Students during the Early COVID-19 Pandemic (*n* = 301).

	Total	Theme 1		Theme 2		Theme 3		Theme 4		Theme 5		Theme 6	
		Yes	No	Yes	No	Yes	No	Yes	No	Yes	No	Yes	No
	*n* = 301	*n* = 96	*n* = 200	*n* = 143	*n* = 153	*n* = 230	*n* = 66	*n* = 263	*n* = 33	*n* = 176	*n* = 120	*n* = 19	*n* = 277
Total vegetables	3.3 ± 0.1	3.2 ± 0.1	3.4 ± 0.2	3.4 ± 0.1	3.2 ± 0.2	3.3 ± 0.1	3.1 ± 0.2	3.3 ± 0.1	3.5 ± 0.3	3.2 ± 0.1	3.4 ± 0.2	3.6 ± 0.3	3.3 ± 0.1
Greens and beans	2 ± 0.1	2.1 ± 0.2	1.9 ± 0.3	2.0 ± 0.2	2.1 ± 0.2	2.0 ± 0.2	2.0 ± 0.3	2.1 ± 0.2 §	1.2 ± 0.4	1.9 ± 0.2	2.2 ± 0.2	1.9 ± 0.6	2.0 ± 0.2
Total fruit	2.1 ± 0.1	2.1 ± 0.2	2.0 ± 0.2	1.9 ± 0.2	2.3 ± 0.2	2.2 ± 0.2	1.8 ± 0.3	2.2 ± 0.1	1.6 ± 0.4	2.4 ± 0.2 **§**	1.7 ± 0.2	3.1 ± 0.6	2.0 ± 0.1
Whole fruit	2.2 ± 0.1	2.2 ± 0.2	2.0 ± 0.2	1.9 ± 0.2	2.4 ± 0.2	2.3 ± 0.2	1.8 ± 0.3	2.2 ± 0.2	1.7 ± 0.4	2.4 ± 0.2	1.8 ± 0.2	2.6 ± 0.6	2.1 ± 0.2
Whole grain	3.1 ± 0.2	3.0 ± 0.3	3.2 ± 0.4	3.3 ± 0.4	2.8 ± 0.3	3.2 ± 0.3	2.8 ± 0.6	3.3 ± 0.3 **¥**	1.7 ± 0.5	3.2 ± 0.3	2.9 ± 0.4	2.9 ± 1.0	3.1 ± 0.2
Total dairy	4.6 ± 0.2	4.9 ± 0.3	4.1 ± 0.4	4.5 ± 0.4	4.7 ± 0.3	4.6 ± 0.3	4.5 ± 0.5	4.7 ± 0.3	3.6 ± 0.7	4.4 ± 0.3	4.9 ± 0.4	6.4 ± 0.9 **§**	4.5 ± 0.3
Total protein	4.2 ± 0.1	4.3 ± 0.1	4.2 ± 0.2	4.3 ± 0.1	4.2 ± 0.1	4.2 ± 0.1	4.5 ± 0.1	4.3 ± 0.1	4.3 ± 0.3	4.2 ± 0.1	4.4 ± 0.1	4.7 ± 0.1 **¥**	4.2 ± 0.1
Seafood and plant protein	2.8 ± 0.2	2.9 ± 0.2	2.8 ± 0.3	2.7 ± 0.2	3.0 ± 0.2	2.8 ± 0.2	3.0 ± 0.3	3.0 ± 0.2	2.0 ± 0.5	2.9 ± 0.2	2.8 ± 0.2	3.4 ± 0.5	2.8 ± 0.2
Fatty acid ratio	5.9 ± 0.2	5.7 ± 0.3	6.2 ± 0.5	5.6 ± 0.4	6.1 ± 0.3	6.1 ± 0.3	5.1 ± 0.5	5.8 ± 0.3	6.4 ± 0.7	5.8 ± 0.3	6.1 ± 0.4	5.2 ± 0.8	5.9 ± 0.3
Sodium	3.8 ± 0.2	3.8 ± 0.3	3.7 ± 0.4	3.6 ± 0.3	4.0 ± 0.3	3.9 ± 0.3	3.3 ± 0.5	3.8 ± 0.2	3.1 ± 0.5	4.1 ± 0.3	3.3 ± 0.3	4.2 ± 0.9	3.7 ± 0.2
Refined grains	6.1 ± 0.3	6.1 ± 0.3	5.9 ± 0.5	5.6 ± 0.4	6.5 ± 0.3	6.2 ± 0.3	5.6 ± 0.6	6.0 ± 0.3	6.8 ± 0.6	5.8 ± 0.3	6.4 ± 0.4	6.5 ± 0.8	6.0 ± 0.3
Saturated fatty acids	5.8 ± 0.2	5.8 ± 0.3	5.7 ± 0.4	5.6 ± 0.3	5.9 ± 0.3	6.0 ± 0.3	5.2 ± 0.5	5.7 ± 0.2	6.1 ± 0.6	5.8 ± 0.3	5.8 ± 0.4	4.7 ± 1.0	5.8 ± 0.2
Added sugar	8 ± 0.2	8.2 ± 0.2	8.0 ± 0.3	8.0 ± 0.2	8.2 ± 0.2	8.2 ± 0.2	7.8 ± 0.4	8.1 ± 0.2	8.5 ± 0.4	8.2 ± 0.2	8.0 ± 0.3	8.4 ± 0.5	8.1 ± 0.2
Total HEI score ^2^	53.8 ± 0.9	54.3 ± 1.0	53.3 ± 1.7	52.5 ± 1.4	55.5 ± 1.1	55.1 ± 1.0 **§**	50.4 ± 1.8	54.4 ± 1.0	50.6 ± 2.1	54.1 ± 1.2	53.7 ± 1.3	57.6 ± 3.3	53.7 ± 0.9

^1^ Theme 1 = “No effects of COVID”; Theme 2 = “Changes in financial ability to buy food”; Theme 3 = “Changes in shopping habits”; Theme 4 = “Changes in food preparation and food or drink consumption”; Theme 5 = “Changes in attitudes, feelings, and habits related to food or eating”; Theme 6 = “Increased cleaning”. ^2^ HEI Score: Healthy Eating Index Score 2015. Independent samples *t*-tests were used to compare HEI 2015 and component scores within thematic categories: “Yes” versus “No”. Symbols indicate differences between thematic codes within themes (“Yes” versus “No”) § indicates *p*-value < 0.05; ¥ indicates *p*-value < 0.01.

## Data Availability

The data presented in this study are available on request from the corresponding author. The data are not publicly available due to privacy issues.

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
