# Peer review of "Influence of COVID-19 Pandemic Restrictions on College Students’ Dietary Quality and Experience of the Food Environment"

_nutrients, 2021, doi:10.3390/nu13082790_

Round 1

Reviewer 1 Report

This is an interesting study on the impact of the first restrictions due to the COVID 19 pandemic on the eating behavior of a sample of university students, mostly women.

The study design is cross-sectional and collects data using qualitative and quantitative methodology on different aspects related to food consumption and eating behavior.

The tools used to analyze the quality of the diet have been validated for the US population.

The results obtained are interesting. Although it is not a representative sample and the study design is cross-sectional, it provides very useful data for comparison with other studies. Some results coincide with those of other investigations carried out with university students. The results on HEI and situations of food insecurity are also very interesting, 35% is a high figure.

The results presented are therefore very interesting, although they would be more comprehensible if more tables were added with the results of the statistical tests, instead of explaining them in the text.

Below, I propose a series of suggestions that in my opinion would improve the presentation of the results.

Format

Put tables 2 and 3 vertically instead of horizontally. It makes the article easier to read and takes up fewer pages.

In addition, the percentages that are cited later in the text could be added for each item and the results of the statistical tests. The information would be more summarized and clearer.

Comments to authors

Lines 173-182. No effects of COVID-19 Pandemic.

It would be convenient to put a table with these results and also explain what statistical test has been carried out, instead of putting only the p-value.

Lines 183-206. Changes in Financial Ability to buy Food

The same for these results.

Lines 274. Changes in attitudes, feelings and habits related to foord of eating.

The same for these results.

It would be interesting to add results for the different ethnic groups that make up the sample.

Author Response

This is an interesting study on the impact of the first restrictions due to the COVID 19 pandemic on the eating behavior of a sample of university students, mostly women.

The study design is cross-sectional and collects data using qualitative and quantitative methodology on different aspects related to food consumption and eating behavior.

The tools used to analyze the quality of the diet have been validated for the US population.

The results obtained are interesting. Although it is not a representative sample and the study design is cross-sectional, it provides very useful data for comparison with other studies. Some results coincide with those of other investigations carried out with university students. The results on HEI and situations of food insecurity are also very interesting, 35% is a high figure.

Thank you. We agree. 35% is in line with food insecurity at many universities in the US, but we agree it is a high figure, too high.

The results presented are therefore very interesting, although they would be more comprehensible if more tables were added with the results of the statistical tests, instead of explaining them in the text.

We are not sure that we can add tables to this manuscript, but we have tried to respond to your suggestions below, adding information as requested.

Below, I propose a series of suggestions that in my opinion would improve the presentation of the results.

Format

Put tables 2 and 3 vertically instead of horizontally. It makes the article easier to read and takes up fewer pages.

We have made this change. Thank you for the suggestion.

In addition, the percentages that are cited later in the text could be added for each item and the results of the statistical tests. The information would be more summarized and clearer.

We have added the n for number and percentage of participants providing responses within a code or theme to Table 2; however, this is not possible for the qualitative data in Table 3. (Note, it is not typical to include numbers and percentages of responses in tables providing qualitative data in part because these were open-ended responses. That is, others may have endorsed these views if asked specifically. Instead, we were trying to understand the overall experience students had and identify key themes. However, we agree this may make it easier for readers to understand how many students felt an issue was significant enough to report)

Comments to authors

Lines 173-182. No effects of COVID-19 Pandemic.

It would be convenient to put a table with these results and also explain what statistical test has been carried out, instead of putting only the p-value.

These results just explain in more detail the numbers endorsing the codes, sub-themes, and themes in Table 2. We have included the number of responses within each of the above on Table 2 as requested. We also indicated which statistical test was used next to the p-value. We apologize for any difficulty in understanding these data. Our intent was to include the qualitative data together with an analysis of how the quantitative data integrate with it.

Lines 183-206. Changes in Financial Ability to buy Food

The same for these results.

See our response above regarding this request.

Lines 274. Changes in attitudes, feelings and habits related to foord of eating.

The same for these results.

See our response above regarding this request.

It would be interesting to add results for the different ethnic groups that make up the sample.

The significant results by race/ethnicity are reported in the manuscript. However, due to space limitations, we don’t think we can split this qualitative data up by race/ethnicity. We appreciate the suggestion. 

Reviewer 2 Report

Thank you for the opportunity to review this manuscript. In general, it is well written. I have the following comments to strengthen and clarify this manuscript:

Lines 118-129:  Coding of qualitative data was conducted by three researchers. During the final step of reviewing codes, how many responses were coded the same by all three researchers and how many responses were coded differently? How were they resolved before finalizing? Adding inter-coder agreement or any other information on this step needs to be included.

Pages 5-9, Table 2: For the order of listing notes, the alphabetical order is not the most helpful nor easiest to compare some notes. For example, contrasting notes such as "Increased fruit and vegetable intake" and "Reduced fruit and vegetable intake" are helpful to have them next to each other. Also, it may make sense to order by the higher to lower frequency of notes reported by participants.

Page 6, Table 2: For the “Changes in shopping habits” theme, please fill in the blank in "sub-theme". If not named and the ones that do not fit in the above sub-themes are included, "others" would be helpful.

Page 7, Table 2: For the “Cooking” theme, it is helpful to include the difference between "cooking less" vs cooking less to avoid food waste. For example, is the former for any reason other than to avoid food waste? Or, is the reason not mentioned?

Lines 363-364, 392-393, 402, 414 and 434: For an easier comparison, please provide both the number and percentages of participants.

Author Response

Lines 118-129:  Coding of qualitative data was conducted by three researchers. During the final step of reviewing codes, how many responses were coded the same by all three researchers and how many responses were coded differently? How were they resolved before finalizing? Adding inter-coder agreement or any other information on this step needs to be included.

As indicated in the Methods lines 120-124, 50 participants’ data was coded in common, iteratively, while the 3 coders communicated. We have edited this to make it more explicitlyclear in line 124. Because nVivo teams was not used, comparing all three coders responses to determine an overall kappa was not possible. Instead, we have tried to provide additional information in lines 126-134 and 136-139 on how the coders guarded against bias and ensured the reliability of the data.

Pages 5-9, Table 2: For the order of listing notes, the alphabetical order is not the most helpful nor easiest to compare some notes. For example, contrasting notes such as "Increased fruit and vegetable intake" and "Reduced fruit and vegetable intake" are helpful to have them next to each other. Also, it may make sense to order by the higher to lower frequency of notes reported by participants.

We have reordered them according to frequency as suggested.

Page 6, Table 2: For the “Changes in shopping habits” theme, please fill in the blank in "sub-theme". If not named and the ones that do not fit in the above sub-themes are included, "others" would be helpful.

We have added “others” for the codes that do not have sub-themes.

Page 7, Table 2: For the “Cooking” theme, it is helpful to include the difference between "cooking less" vs cooking less to avoid food waste. For example, is the former for any reason other than to avoid food waste? Or, is the reason not mentioned?

We have added numbered footnotes to explain these codes in cooking and similar codes in Shopping. 

Lines 363-364, 392-393, 402, 414 and 434: For an easier comparison, please provide both the number and percentages of participants.

This has been added.